# Impacts of Dietary Nutritional Composition on Larval Development and Adult Body Composition in the Yellow Fever Mosquito (*Aedes aegypti*)

**DOI:** 10.3390/insects11080535

**Published:** 2020-08-15

**Authors:** Tess van Schoor, Erin Taylor Kelly, Nicole Tam, Geoffrey Michael Attardo

**Affiliations:** Department of Entomology and Nematology, University of California, Davis, Davis, CA 95616, USA; tvanschoor@ucdavis.edu (T.v.S.); etkelly@ucdavis.edu (E.T.K.); nbtam@ucdavis.edu (N.T.)

**Keywords:** *Ae. aegypti*, nutrition, larval development, diet, pupation, carbohydrates, protein, lipids

## Abstract

**Simple Summary:**

The mosquito *Aedes aegypti* (*Ae. aegypti*) is responsible for the spread of viruses such as Zika and Dengue. The nutritional environment of immature *Ae. aegypti* is important for development of larvae and resulting adult mosquitoes. Larval mosquitoes with inadequate nutrition can result in developmental failure or impact the size and reproductive ability of adults. Understanding the nutritional requirements of larval mosquitoes allows us to optimize lab reared mosquitoes and identify new targets for mosquito control. We tested the effect of diets with different ratios of protein to carbohydrates on the life history traits of *Ae. aegypti*. Each diet was composed of autolyzed Brewer’s yeast (protein), and/or rice flour (carbohydrates). Larvae fed a medium-low protein diet had the shortest pupation time. As adults, the medium-low protein dietary group also had the longest wing lengths, highest weights, and increased lipid stores compared to the adults in all other dietary groups. These findings indicate that both carbohydrates and protein are essential components of *Aedes aegypti* larval diets. However, *Ae. aegypti* larvae fed a diet rich in carbohydrates and lower in protein seem to flourish as long as they receive enough dietary protein to fulfill basic biochemical requirements for growth and development.

**Abstract:**

Background: the mosquito *Aedes aegypti* (*Ae. aegypti*) is an important vector of arboviruses, including Zika, Dengue, and Chikungunya. The dietary requirements of larval *Ae. aegypti* are not well understood and likely impact developmental and physiological parameters knowledge of which could be important for vector control. This study examines the effects nutrition has on growth and development of larval *Ae. aegypti* of laboratory-reared Rockefeller strain mosquitoes. Methods: mosquito larvae were split into five feeding groups with diets providing different ratios of protein and carbohydrates. Each group received autolyzed Brewer’s yeast (AY - high-protein), and/or rice flour (RF—high-carbohydrate). The groups were monitored to record larval developmental times, adult sizes and nutritional stores. Results: the 100% AY group failed to pupate, suggesting the AY alone is either lacking in critical nutrients or is toxic at higher concentrations. The 100% RF group resulted in the smallest adults that took the longest time to reach pupation. Of the remaining groups, the 25% AY/75% RF (Med–low) diet yielded adult mosquitoes with highest average weight, wing length, and lipid stores relative to the other diets. Conclusions: the dietary requirements for development, body size, and nutrient stores of *Ae. aegypti* mosquitoes appear to be dependent on a relatively low but essential proportion of dietary protein to carbohydrates to achieve optimal developmental outcomes.

## 1. Introduction

The Yellow Fever mosquito, *Aedes aegypti* (*Ae. aegypti*), is responsible for the transmission of an array of arboviruses that are responsible for tremendous morbidity and economic loss. Currently, the CDC recommends aerial spraying of both Naled, an organophosphate adulticide, and *Bacillus thuringiensis* subspecies *israelensis* (Bti), a bacteria toxic to larvae, as the primary means for *Ae. aegypti* population control [1]. However, the implementation of sterile insect technique (SIT) based methods are appealing alternatives due to the widespread prevalence of insecticide resistance in *Ae. aegypti* populations across the globe [2,3,4].

This strategy in combination with other methods can reduce the overall reproductive rate of a population below that required for maintenance resulting in destabilization and ideally the crash of a population [3]. The effective implementation of SIT requires the release of large quantities of lab-reared, sterilized male *Ae. aegypti* for competition with wild, fertile males for females. However, the process of mass rearing these mosquitoes requires maintenance under optimal conditions to optimize developmental time and fitness of resulting adults. Larval nutrition is an important consideration in evaluating the optimum rearing conditions for male *Ae. aegypti* to utilize SIT based strategies as a means of vector control via the disruption of reproduction [5]. In addition, *Ae. aegypti* is frequently used as a model system in which to study basic mosquito physiology and vector pathogen interactions. However, the dietary regimens for larval mosquito rearing vary significantly between labs in terms of the type of foods and amounts used in rearing protocols. Examples of laboratory diets include fish food (typically Tetramin brand), liver powder, and various rat and cat chow varieties [5,6,7].

The impact of larval diet composition on fitness parameters, such as immature development time, adult body size, and stored nutrient composition, remains vague. Understanding of the requirements and roles individual classes of nutrients play in regulating larval development is important for practical purposes, including the development of rearing protocols to enhance experimental replicability and advance efficient mass rearing practices. Analysis of larval nutrition and the resulting impact on growth and maturation facilitates a deeper understanding of the biochemical requirements and pathways mediating the developmental processes in immature mosquitoes. The target of rapamycin (TOR) and insulin/phosphoinositide 3 kinase (PI3K) signaling cascades are conserved and well characterized mediators of growth and development in most metazoan organisms [6,8,9]. Knowledge of these aspects of mosquito biology can provide insights into exploitable physiological processes and the effects of larval nutrition on vectorial capacity. Multiple environmental factors can impact the rate and success of mosquito larval development, including temperature, larval density, diet, and environmental microbiota [10,11]. The lack of adequate nutrition during larval development of *Ae. aegypti*, can result in delayed or failed development [12,13] or in production of adults lacking suitable nutritional stores [6]. As a result, these mosquitoes often require multiple blood meals prior to reproduction, which results in increased vector/host interactions and vectorial capacity [14,15,16]. Mosquito larvae require sugars, nucleotides, polyunsaturated fatty acids, sterols, vitamins, and fourteen essential amino acids for proper development [12,17,18,19]. Additionally, recent work by Coon et al., reported that *Ae. aegypti* are dependent on gut microbiota to assimilate nutrients for growth and development [20]. E. coli perform aerobic respiration using cytochrome *bd* oxidases, which stimulate larval molting. *Ae. aegypti* larvae with non-functional cytochrome *bd* oxidases exhibit poor growth [20]. This study aims to assess the impact of dietary composition, in terms of how the relative proportions of carbohydrates and proteins affect larval development by rearing *Ae. aegypti* larvae on diets with differing ratios of these nutrients. The developmental effects were monitored via developmental and physiological parameters, including larval developmental time, pupation rate, adult body size/weight, and adult nutritional reserves.

## 2. Materials and Methods

### 2.1. Diets

Five diets were compared for this study (Figure 1). Each diet is composed of a combination of Yeast Hydrosylate Enzymatic Autolyzed Yeast (AY) (MP Biomedicals, Burlingame, CA, USA) and/or rice flour (RF) (Erawan Group, Bangkok, Thailand). The diets were formulated with different ratios of these two ingredients. Autolyzed Brewer’s yeast extract, the water-soluble component of autolyzed Brewer’s yeast, is composed of *Saccharomyces cerevisiae* grown on a sugar-rich medium, such as beet molasses [21,22]. The relative proportion of nutrients in AY is 60% protein, 31% carbohydrates, and 9% lipids. The other ingredient, RF, has a nutritional constitution of 6.6% protein, 1.4% lipids, and 92% carbohydrates [23]. The five diets studied were 100% RF (Low protein), 25% AY and 75% RF (Med–low protein), 50% AY and 50% RF (Med protein), 75% AY and 25% RF (Med–high protein), and 100% AY (High protein). Figure 1 visualizes the relative nutritional proportions of these diets. Larvae for these experiments were synchronously hatched under vacuum to ensure they were equivalent in age and developmental status. On the day of hatching, larvae were fed 0.31 mL of diet. On day 1, larvae were fed 0.10 mL of diet. On day 2, larvae were fed 0.20 mL of diet. On day 3, larvae were fed 0.30 mL of diet. On day 4, larvae were fed 0.60 mL of diet. On day 5, the larvae were fed 0.31 mL of diet. On day 6 and each day after initiation of pupation, residual fourth-instar larvae were provided 0.15 mL of diet to avoid over-feeding and fouling the water. The larvae consumed most if not all of the diet each day based on visual inspection. The larvae were fed at 3:00 PM every day.

### 2.2. Effects of Diets on Larval Development

For this study, the lab-based Rockefeller strain of *Aedes aegypti* was used. In our laboratory, this strain is typically maintained on a larval diet of Tetramin fish food. Groups of 200 first-instar *Ae. aegypti* larvae were counted and placed in 34.3 × 25.4 × 3.8 cm Bioquip 1426B plastic trays (Bioquip Products, Inc., Compton, CA, USA) containing 600 mL of dechlorinated tap water. The tray water level was checked daily and water was added as needed to maintain 600 mL of volume. Larvae were reared at 26 °C and 80% ambient humidity. Each diet is represented by three trays with each tray representing a biological replicate including 200 larvae. The number of larvae reaching pupation were counted daily for each tray upon initiation of pupation in any of the dietary treatments. Pupal counts were performed daily over six days to determine the mean pupation time. The total number of larvae reaching pupation was divided by the starting number of larvae per tray to determine the proportion reaching pupation.

### 2.3. Adult Body Size

Mosquitoes were frozen and stored at −20 °C for 24 h. The bodies were then placed in a 50 °C drying oven for 24 h to eliminate excess body moisture. After drying, adult females and males were stored in petri dishes at room temperature, 20 °C. Fifteen females and fifteen males were chosen for body weight and wing length measurements from each treatment replicate. Each adult’s dry weight was measured on a Mettler balance (Mettler Toledo, Columbus, OH, USA). The adult body size was represented by a measurement of the right and left wings from the distal edge of the alula to the end of the radial vein (excluding fringe scales). The wings were removed and taped to a microscope slide. Wing lengths were measured using a Dino-Lite microscope (Dino-Lite Scopes, Torrance, CA, USA), 1.75 cm from mount to lens, while using a magnification of 40×.

### 2.4. Nutritional Analysis

Fifteen adult males and fifteen adult females from each treatment replicate were collected 24 h post-eclosion. These individuals were then frozen and stored at −80 °C. Sugar, glycogen, and lipids were extracted from each individual mosquito using methods that were adapted from Van Handel and Day [24,25]. Individual dried mosquitoes were homogenized and extracted in 0.5 mL of chloroform/methanol (1:1) in 16 × 100 mm borosilicate glass test tubes. The homogenate was vortexed for 15 s and then centrifuged for 1 min. to separate the aqueous from the organic phase. The aqueous phase containing sugar and glycogen was moved to another test tube and 0.1 mL of 2% sodium sulfate was added followed by vortexing and centrifugation to precipitate the glycogen. The supernatant containing simple sugars was moved to another tube. The remaining liquid in the test tubes was allowed to evaporate off in a 95 °C heating block until the tubes were dry. Anthrone reagent (for detection of carbohydrates) was created by slowly pouring 150 mL of ddH2O into 380 mL of concentration sulfuric acid followed by addition of 750 mg of anthrone. Vanillin reagent (for detection of lipids) was created by dissolving 600 mg vanillin in 100 mL of hot water followed by the addition of 400 mL of 85% phosphoric acid. Sugar and glycogen samples were treated with 1 mL of anthrone reagent followed by heating at 95 °C for 17 min. After heating, the sugar and glycogen samples were vortexed and sample absorbances were read at 625 nm on a NanoDrop spectrophotometer (ThermoFisher, Waltham, MA, USA). Dried lipid samples were treated with 0.2 mL of concentrated sulfuric acid and heated at 95 °C for 10 min. After heating, 1 mL of vanillin reagent was added to the tubes, which were then vortexed and allowed to incubate for 10 min. at room temperature. After incubation, the sample absorbances were read at 525 nm on a NanoDrop spectrophotometer. Standard curves were generated via serial dilutions of stock solutions of 1 mg/mL anhydrous glucose in water for glycogen and sugar analysis and 1 mg/mL Peanut oil in Chloroform for lipid analysis.

### 2.5. Statistical Analyses

Data were analyzed using R Version 3.6.2 in RStudio Version 1.2.5033 software for Macintosh (R Foundation for Statistical Computing, Vienna, Austria). Curves for pupation rates for each diet were generated by generalized linear model analysis in R. The grey bands around the curves represent the 95% confidence interval. The “survival” package was used to perform Kaplan–Meier survival analysis followed by pairwise statistical comparison of the experimental groups by log-rank in the “survival” package [26,27]. Physiological data was tested for normality using the Shapiro–Wilks test. The results indicated the datasets were likely non-normal and non-parametric statistical tests were chosen for further analyses. Comparisons of physiological features (mean weights, wing lengths and nutrients) for each diet were performed by Kruskall-Wallis test followed by pairwise scoring using the Wilcoxon signed-rank test for each sex. Packages used for figure production and statistical significance annotation include “ggplot2” [28], “aod”, “ggfortify” [29], “forcats” [30], “dplyr” [31], “ggthemes” [32], “ranger” [33], “survminer” [34], “multcompLetters4“ [35], “RVAideMemoire” [36] and “ggpubr” [37] (Code and raw data used for analyses are provided in Appendix A and in Appendix A). Tables summarizing *p*-values for all comparisons are provided in Appendix A. A threshold of *p* < 0.05 was chosen to indicate statistical significance for all of the analyses. 

## 3. Results

### 3.1. Dietary Effects on Larval Development

The larval time to pupation was tracked for each treatment to measure the effect of diet on developmental rate of the larvae (Figure 2A,B). Of note, larvae that were reared on the High protein diet (AY only) showed high levels of mortality and surviving larvae failed to pupate in the six days following initiation of pupation by other groups at which point the experiment was terminated. Due to the inability to collect adults from this group, they are not included in subsequent analyses. The remaining treatments all pupated; however, significant differences were observed in the time to pupation between treatments. The dietary group with the slowest median time to pupation was the Low protein (Low) diet (nine days) (Figure 2A). The mosquitoes on the Medium–Low protein (Med–low), Medium protein (Med), and Medium-high protein (Med–high) diets had a median time to pupation of ~7 days with the Med–high diet lagging slightly behind Med–low and Medium diets. Statistical comparison of larval development rates across the diets reveals that larvae on the Med–low protein and Medium protein diets underwent equivalent rates of development. Larvae on the Med–high protein diet has a significant lag behind the Med–low and Med diets. Mosquitoes that were reared on the Low protein diet (RF only) had the slowest rate of development relative to diets Med–low protein, Medium protein, and Med–high protein (Figure 2A,B).

### 3.2. Dietary Effects on Adult Body Size

The quantification of the effect of dietary constitution on adult size was performed via measurements of dry weight and wing length (Figure 3A,B—*p*-value matrices are available in Appendix A). Dietary factors had a significant effect on adult body size in both male and female mosquitoes across treatments (Figure 3A,B). None of the individuals in the treatment group receiving the High protein diet survived to pupation and are not included in the adult body size analysis. A comparison of the body weights and wing lengths of mosquitoes from the other four treatment groups revealed that the diets had significant impacts on both dry weight and wing lengths. The most dramatic impact was on the population receiving rice flour alone. These mosquitoes were significantly smaller relative to all of those with some autolyzed yeast in their diets in both male and female groups. The groups receiving the Med–low diet were consistently the largest in both sexes. The diets containing more than 25% AY (Med and Med–high) showed significant decreases in overall body size in both sexes, as reflected by both dry weight and wing lengths. The observed size reductions appear to negatively correlate with protein concentration. This effect was observed in both sexes.

In females, significant differences in dry weight were observed between all four diets (*p*-values < 0.001). The Med–low protein diet produced the largest individuals with a mean weight 340 mg greater than that of the Low protein diet which had the lowest average weight (Figure 3A). Of interest is that the dry weights of females from the higher protein Med and Med–high diets were significantly reduced in mean weights relative to the Med–low protein diet. Male dry weights behaved in a similar manner to the females, with the exception that a significant reduction in dry weight was not observed between the Med–low and Med diets. Similar to females, the largest difference in adult male body sizes was observed between the Low and Med–low protein diets, with individuals from the Med–low diet on average 190 mg heavier than the RF only Low group (*p*-value < 0.001) (Figure 3A). Males from the Med–high protein diet also showed a significant reduction in mean weight (37.11 mg) relative to males fed the Med–low and Med diets (*p*-values < 0.001 and < 0.03, respectively) (Figure 3A).

The wing lengths of both males and females followed almost identical trends as the dry weight measurements with statistically significant differences observed between all diets for both sexes (Figure 3B). As observed for the dry weights, mosquitoes of both sexes on the Med–low diet had the longest wing lengths and those on the Low diet the shortest. Statistical comparisons between all groups for both sexes were significant (*p*-values < 0.03). The negative correlation between increasing dietary protein levels and reduced size is also observed in the wing length data (Figure 3B).

### 3.3. Nutritional Stores

Analysis of nutrient stores in individuals revealed significant differences in simple sugar, glycogen, and lipid levels across the different diets. Analysis of differences in sugar content revealed that adult females reared on the higher carbohydrate Low and Med–low diets had significantly higher levels of sugars relative to those in the higher protein Med and Med–high diets (*p*-values < 0.02). In males, no significant differences were observed in sugar content between the groups (Figure 4A, *p*-value matrices available in Appendix A).

Glycogen stores varied significantly relative to both diet and sex with higher protein diets correlating with higher glycogen levels (Figure 4B). Female mosquitoes from the higher protein Med and Med–high diets had significantly higher glycogen stores relative to those in the Low and Med–low protein groups (*p*-values < 0.02) (Figure 4B). A comparison of glycogen stores per diet for males revealed a similar trend as that of females. The glycogen stores were highest in males reared on the Med diet and were significantly higher than those from all other diets (Low (*p*-value < 0.0001), Med–low (*p*-value < 0.0001), Med–high (*p*-value < 0.009)) (Figure 4B). The male Med–high group had the second highest level of glycogen was also significantly higher than males from the lower protein Low and Med–low dietary groups (*p*-values < 0.003). The Low diet generated males with the lowest stored glycogen levels.

Lipid stores also varied significantly by diet in both sexes (Figure 4C). As opposed to glycogen, lipid levels were highest in females reared on the Low protein diet and were significantly higher than the Med (*p*-value < 0.001) and Med–high (*p*-value < 0.001) diets,, but the difference between the Med–low diet was not significant (Figure 4C). Adult females on Low protein diet retained an average of 54.07 µg more lipid stores than those with this lowest lipid stores on the high protein Med–high diet. The female data shows a downward trend in lipid stores in diets with higher concentrations of protein. Males from the Med–low diet had the highest levels of stored lipids and were significantly higher than in males fed the Med (*p*-value< 0.001) or Med–high diets (*p*-value < 0.02) (Figure 4C).

## 4. Discussion

### 4.1. Larval Developmental Time Is Shortest and Adult Body Sizes Are Largest on a Diet High in Carbohydrates with a Low Proportion of Protein

Larval developmental time is dependent on a properly balanced diet, and the shortest development times and largest adult body sizes were achieved with a high proportion of RF and a relatively low proportion of AY (Med–low protein diet). However, in the absence of AY, larval growth is stunted in terms of both developmental rate and body size as mosquitos on RF only were the slowest to develop and had the smallest adult body sizes. This suggests that a component of AY, possibly the high protein content, is important for larval growth. However, too much of this component appears to slow or stunt growth as diets with higher proportions were reduced in body size and were somewhat slower to develop. Additionally, pure AY inhibited development completely, suggesting that, at increased concentrations, it becomes toxic or the mosquitoes are not getting enough of a nutrient derived from the RF. It is possible that the baseline levels of carbohydrates could be a limiting factor for successful development. Female body sizes were more dramatically impacted by the diet treatments than males. Female *Aedes aegypti* are known to readily generate lipids from carbohydrates, which can explain their ability to grow to large sizes on the Med–low protein diet [38]. A significant observation from this study is that the female body sizes were impacted in a more dramatic manner in response to different diets relative to males. This may result from higher female dietary requirements due to increased nutrient baseline requirements tied to energetically expensive reproductive processes such as vitellogenesis and oogenesis [39]. We also speculate that the female mosquitoes may be particularly sensitive to larval dietary protein requirements, as has been observed in *Drosophila* [40,41,42].

This observation might be associated with TOR signaling, as, without the proper amino acid signals, the TOR pathway will not be activated and would likely negatively affect growth and development. Nutritional signaling via the TOR pathway is documented as an important regulator of growth and development across animals from many taxa, including *Ae. aegypti* [43,44,45]. The larval fat body plays a critical role in regulating hormone and nutrient levels throughout the insect [16]. As the TOR pathway requires certain amino acids to activate growth associated signaling cascades that coordinate larval growth, inadequate dietary amino acids may be preventing the secretion of growth factors from the fat body, resulting in delayed larval growth and development [42]. This process has been observed during larval development in *D. melanogaster* [43,46] and nutritional conditions during larvigenesis are shown to impact the secretion of insulin-like peptides in adult *Ae. aegypti* [6]. The TOR signaling cascade also functions to inhibit autophagy (the salvage of nutrients from cellular components) [47,48]. In this manner, dietary protein and active TOR signaling are likely important for the maximization of larval growth and development. This was observed in *C. elegans* where mutations in genes involved in the TOR signaling pathway result in larval developmental arrest [49]. Similarly, *Drosophila* insulin-like peptides (DILPs) promote growth by binding to insulin receptors of target tissues and activating a phosphoinositide 3 kinase (PI3K) signaling cascade to stimulate growth by inhibiting a specific transcription factor, dFOXO [8,9]. In *D. melanogaster*, hyperactivation of the PI3K pathway results in premature larval wandering, which, under normal conditions, only occurs when the larva is preparing to pupate [9]. This suggests that larval behavior and developmental dynamics are, at least in part, being mediated by insulin and nutrient based signaling.

### 4.2. Diets with a High Proportion of AY Result in Adults with Higher Glycogen and Lower Stored Lipids

Our observations suggest that *Ae. aegypti* can thrive in environments providing high proportions of carbohydrates, but key nutrients that are derived from a protein source are required for *Ae. aegypti* for a complete diet. Without those nutrients it appears the mosquitoes are not able to capitalize on high levels of dietary carbohydrates. Larva of *Ae. aegypti* have high dietary plasticity, however Souza et al. determined that bacterial and microalgal diets are not optimal for acquisition of adequate nutrient stores by *Ae. aegypti* larvae [11]. These diets lead to slow pupation and low survival rates. However, *Saccharomyces cerevisiae* presents as a viable option for mass-rearing *Ae. aegypti* larvae. After completing nutritional analysis, *S. cerevisiae* was shown to contain similar amounts of carbohydrates and protein as the standard Tetramin diet. However, despite decreased survival and slower development, larvae that were fed with microalgae and bacteria were still able to complete development [11]. Therefore, Souza et al. show that *Ae. aegypti* larvae demonstrate high dietary plasticity.

Algal species *Chlorella* sp. and *A. platensis* contain high amounts of protein when compared to the standard Tetramin diet [11]. The yeasts studied, *S. Cerevisiae and Pseudozyma* sp., are also protein rich. However, because the yeasts contain significantly more carbohydrates than the microalgae, the first instar larvae fed microalgae spent significantly more time in the larval stage than those fed yeast. Therefore, Souza’s results support the theory that carbohydrates are limiting macronutrients for development and pupation.

Glycogen is an important storage molecule for insects as a substantial amount of glycogen must be incorporated before the organism enters diapause [50,51,52]. The adult mosquitoes on the Medium protein and Med–high protein diets showed the highest glycogen concentrations in both males and females. This observation suggests that diets with higher protein composition and lower carbohydrates capacitate the synthesis and storage of glycogen over lipids and vice versa. The mobilization of fat body glycogen and trehalose allows for the organism to survive when nutrients are not readily available in the environment [53]. In mosquitoes, glycogen is frequently used as storage from which carbohydrates required for flight energy can be quickly generated [54]. Lipids often take longer to convert back to a usable source for the flight muscles [53]. In *D. melanogaster*, fat body glycogen can be mobilized in accordance with glucose levels in the hemolymph and in specific tissues [53]. Fat body glycogen synthesis begins during the late larval period [53,55,56].

In males, the predominant stored nutrient is glycogen, which is primarily used for flight energy during mate seeking [57]. In females, lipid stores are often more abundant, as they are required for the nutrient and energy intensive processes of vitellogenesis and oogenesis [57,58]. The shift towards glycogen in both sexes reared on the high AY diets suggests that something in these diets is capacitating glycogen formation and storage over lipids. We speculate that in the context of larval development the insulin pathway may be playing a role in the regulation of lipid biosynthesis and storage. If carbohydrate concentrations are not high enough the insulin pathway may not be stimulated highly enough to induce lipogenesis. The insulin like peptide 3 (ILP3) regulates lipid and carbohydrate storage in adult *Ae. aegypti* in response to sugar feeding. It is possible that ILP3 or an alternative ILP is performing an equivalent function in larvae [59]. Additionally, in *Ae. aegypti*, CRISPR based knockdown of the *ilp2* and *ilp6* genes resulted in smaller body size, delayed development, and dramatically reduced lipid storage [60]. These findings suggest that insulin signaling is playing an important role in regulating of nutrient storage and metabolism as well as development. As a result, we hypothesize that the relative composition of dietary nutrients may impact how dietary nutrients are metabolized and the form in which they are stored.

Another observation from our data that supports this hypothesis is that adult lipid levels are the highest in females on the Low protein diet, even though those mosquitoes have the smallest body size out of all the diets. A possibility is that the high carbohydrates are triggering insulin signaling resulting in a high rate of lipid synthesis and storage [41,42,49,54]. However, the lack of amino acids in the high carb diets results in reduced signaling by the TOR pathway. The low protein diet likely provides access to large amounts of carbohydrates that stimulate insulin signaling which activates lipid synthesis and storage. We hypothesize that the Med–low protein diet provides the best of both worlds with enough protein to stimulate the TOR pathway to induce cellular growth, replication, and organismal development, while carbohydrates facilitate lipid synthesis and storage.

A consideration and potential weakness of this study to keep in mind while interpreting this data is the adaptability of our lab strain, which has been maintained on a diet of Tetramin fish food. It is possible that, in the presence of sustained selective pressure, adaptive turning to this diet could also impact their responses to these diets. It is also important to note that while autolyzed Brewer’s yeast extract is high in protein it also includes various other nutritional components, such as lipids, vitamins, and nucleic acids that are not accounted for by these experiments and likely play important roles in growth and development [21,22]. Future studies are required in order to determine the role that these compounds play in the effects observed here.

### 4.3. High Concentrations of AY Appear to be Toxic Inhibiting Larval Growth and Pupation

The AY only diet killed or stunted all of the mosquitoes reared under that regimen. There are a few hypotheses that can explain this result. One likely possibility was that the high levels of protein resulted in the production of toxic concentrations of ammonia. This is due to the increased levels of nitrogenous waste associated with protein and amino acid metabolism. High levels of ammonia that are produced during protein digestion have been demonstrated to be toxic to *Ae. aegypti* [61,62]. Dietary protein is degraded to amino acids in the mosquito gut lumen and absorbed by the midgut epithelial cells [63,64]. The absorbed amino acids are then transported through hemolymph and available for uptake by the fat body [65]. Ammonia is produced when amino acids are deaminated to form alpha-keto acids. Ammonia toxicity is demonstrated to have a negative impact on the development rates of larvae [61]. Dias, Rodrigues, and Silva observed that development time of *Anopheles darlingi (An. darlingi)* larvae increased significantly with chronic ammonia treatment. In addition, increased ammonia treatment of L1 *An. darlingi*, in both chronic and acute applications, increased larval mortality [62].

Another possible explanation for our results is that carbohydrate acts as a limiting nutrient. In other species low protein to carbohydrate ratios in larval diets yields adults with preferential life history traits in other species. For example, Barragan et al. found that black soldier flies (*Hermetia illucens*) that were fed a larval diet high in carbohydrate retained high amounts of lipid stores as adults [66]. Along these lines, females in the high protein 75% AY diet had the lowest lipid levels of all the diets tested. The high protein diet may bias against the biosynthesis and storage of lipids, as many lipids are synthesized from hexoses [67]. We speculate that larvae fed the pure AY diet may be attempting to compensate by synthesizing other key nutrients from protein, but, as a result, they are generating a toxic amount of nitrogenous waste. The larvae seem to thrive under high carbohydrate conditions as long as they have enough protein presumably to activate crucial growth and developmental processes. If carbohydrates are a limiting factor, the mosquitoes may lack the energy to support basic functions, let alone nutrient storage and are dying of malnutrition. As a result, the poor survival observed in the high AY diets may be due to a combination of incomplete nutrition and ammonia toxicity.

## 5. Conclusions

Diet plays an important role in immature development in all organisms and they can vary significantly based on the life history of the species. This is particularly so for insects, as they rapidly evolve to occupy specialized niches with potential nutritional limitations [68]. Different species of mosquitoes have different life histories and larval habitats. While *Ae. aegypti* originated as tree hole mosquitoes, populations have adapted to man-made container habitats and readily develop in temporary water sources, such as tires, food containers, flowerpots, and planters [69]. These environments are typically composed of a mixture of leaf and animal detritus, and a previous study found that optimal *Aedes aegypti* development relies on high relative nitrogen content [70]. It appears that the dietary needs of *Ae. aegypti* are tuned, such that they thrive in high carbohydrate environments, but must maintain a baseline of protein intake to take advantage of these nutrients. This likely reflects what is found in their natural habitats with carbohydrates reliably available from plant-based materials that are supplemented with nutrients from animal detritus and co-habiting bacteria. Protein can be supplanted later in life via vertebrate blood meals typically required for egg development by anautogenous female mosquitoes. The nutritional dependencies of larval mosquitoes could be an important target for development of alternative larval control measures.

Overall, *Ae. aegypti* larvae fed the Med–low protein diet exhibited the most preferential life history traits. However, the Med–low protein diet may not be the optimal larval diet as there are many combinations of protein to carb dietary ratios that have not been tested, and there are other factors that affect growth besides protein and carbohydrate intake. The next step in this research is to query the molecular/biochemical mechanisms dictating the outcomes of larval development in *Ae. aegypti*. Analysis of the hormonal and nutritional signaling responses to these different dietary regimes will be important for understanding the key triggers and requirements for the critical processes of larval development and pupation in this important disease vector.

## Figures and Tables

**Figure 1 insects-11-00535-f001:**
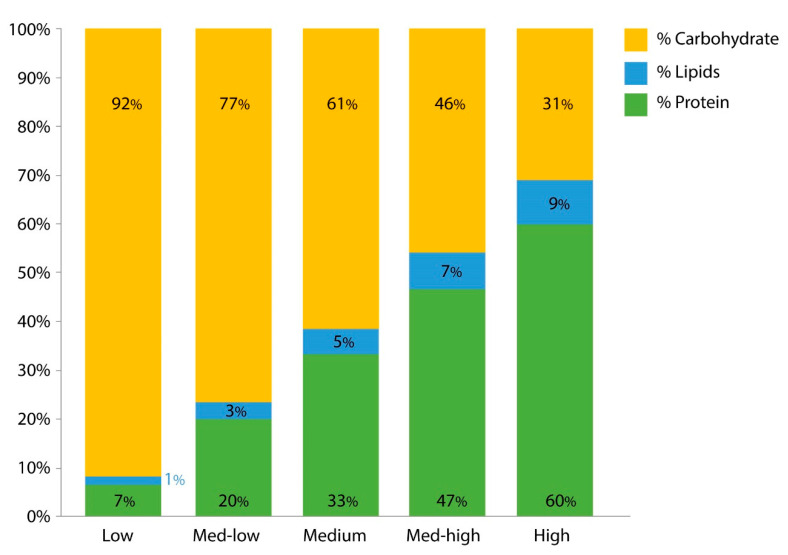
Carbohydrate, fat, and protein composition of the 5 diets composed of varying ratios of autolyzed yeast (AY) and rice flour (RF). The diet ratios are as follows: Low = All RF, Low-Med = 1 part AY: 3 parts RF, Med =1 AY:1 RF, High-med 3 AY:1 RF, High = All AY. Nomenclature represents relative amount of protein associated with each diet.

**Figure 2 insects-11-00535-f002:**
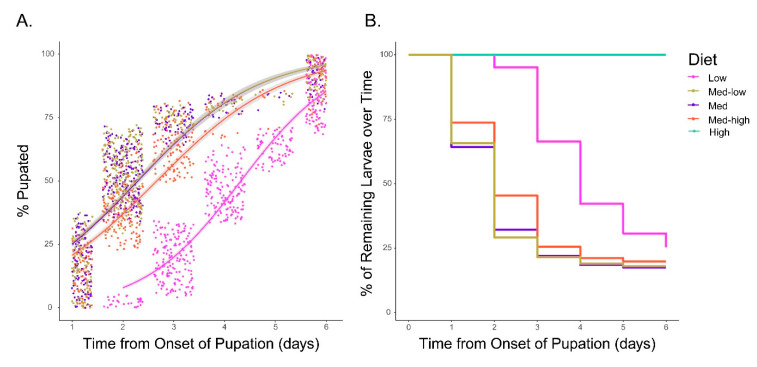
Analysis of pupation rates on different diets. (**A**) Analysis of pupation rate by diet. Points represent the time to pupation for individual mosquitoes within each diet relative to initiation of pupation across all groups. Trend lines represent generalized linear models derived from the data for each diet. Grey areas above and below the trendlines represent 95% confidence intervals. (**B**) Kaplan–Meier Survival analysis shows the percent of remaining larvae for each diet over the six days following onset of pupation. Pairwise comparison by log-rank test of all diets resulted in significant differences between pupation rates (*p* < 0.001) for all comparisons with the exception of the Med–low and Med diets.

**Figure 3 insects-11-00535-f003:**
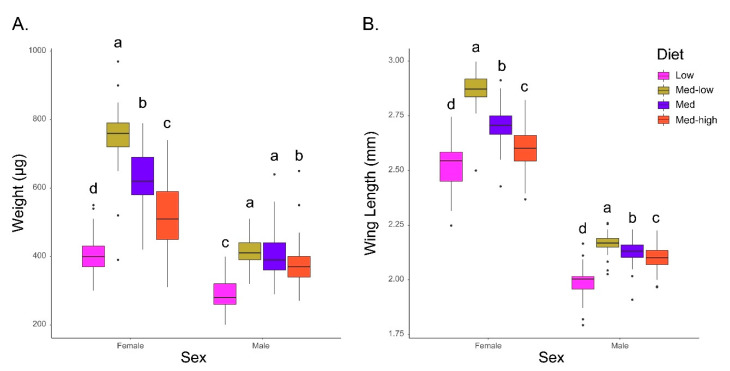
Analysis of dietary impacts on body size. (**A**) Dry body weight in µg. Adults were frozen and then dried for 24 hours before weighing. (**B**) Wing Length in mm. Mosquitoes were measured from alula to the end of the radial vein excluding fringe scales. Letter annotations represent statistical groups (*p* < 0.05) as determined by Kruskall–Wallis test followed by pairwise scoring using the Wilcoxon signed-rank test. Bars sharing letters are not significantly different. Statistical comparisons are only within sex. *p*-value matrices available in Appendix A.

**Figure 4 insects-11-00535-f004:**
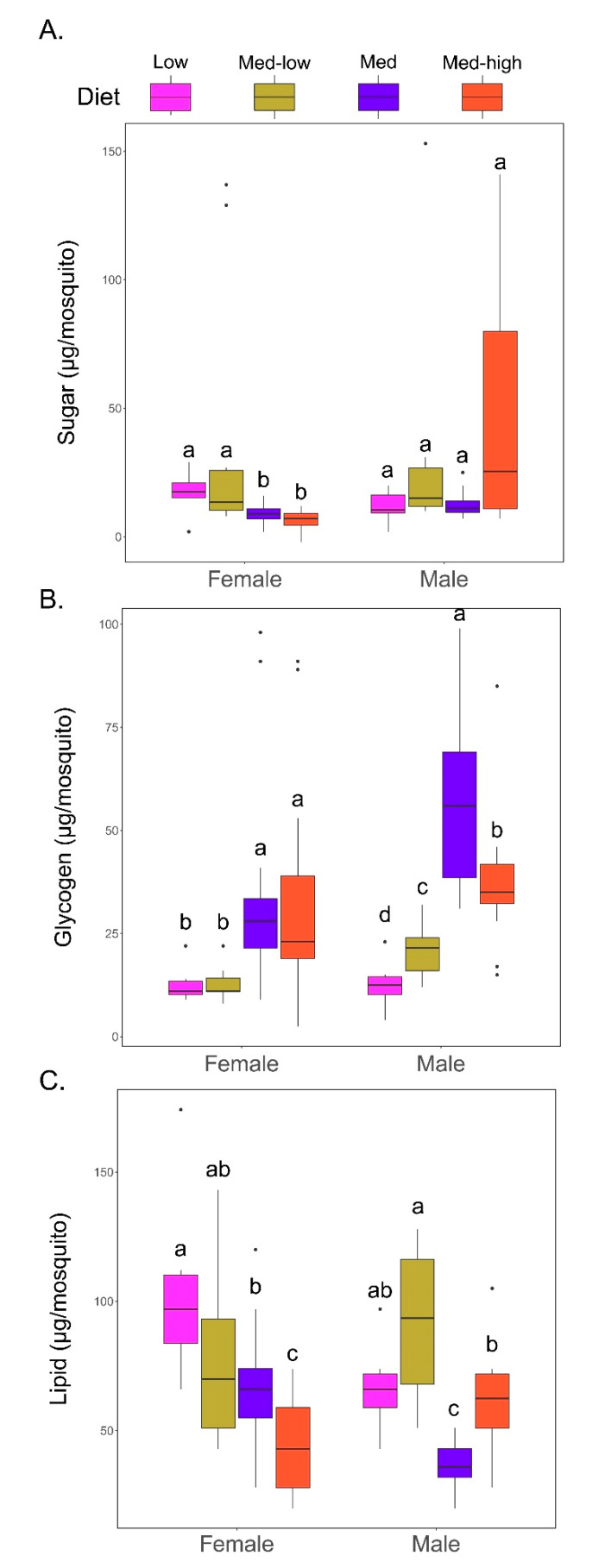
Effect of larval diet on Nutrition Stores of (**A**) Sugar, (**B**) Glycogen, and (**C**) Lipids. Letter annotations represent statistical groups (*p* < 0.05) as determined by Kruskall-Wallis test followed by pairwise scoring using the Wilcoxon signed-rank test for both sexes. Bars sharing letters are not significantly different. Statistical comparisons are only within sex.

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
