# Peer review of "Impacts of Dietary Nutritional Composition on Larval Development and Adult Body Composition in the Yellow Fever Mosquito (Aedes aegypti)"

_insects, 2020, doi:10.3390/insects11080535_

Round 1

Reviewer 1 Report

  1. Brief summary:
    1. The authors present work examining the effects of diets of varying ratios of protein to carbohydrates on male and female larval development time, adult size, and nutritional stores (sugar, glycogen, and lipids). Protein-rich diets alone prevent pupation and high carbohydrate diets alone are likewise developmentally compromised, though to a lesser extent, suggesting carbohydrates as a limiting nutritional requirement. The ideal combination resulted from a low protein (25%), high carbohydrate (75%) diet, perhaps due to protein-stimulated TOR pathway signaling and resulting cellular growth. These findings are of interest to entomologists in general and vector biologists in particular, since they can be used to improve lab rearing protocols, especially those aimed at optimizing fitness of mass reared mosquitoes for use in sterile insect-based mosquito control methods.
  2. Broad comments:
    1. The abstract and introductions are perfectly readable and lucid. The materials and methods section would benefit from some further, relatively minor clarifications (see specific comments). The results would benefit from additional editing to improve readability and to clarify what conclusions the statistical methods and results do and do not support. In general, the denoting of statistical significance in the graphs is confusing, particularly in the cases with double letter labels. Furthermore, some of the labels do not appear to correspond with the write up in the main text of the results. Few statistical test values are presented to in the results (although they are often referred to), which is an omission that needs to be addressed. However, the discussion generally does an excellent job at teasing apart the implications and potential rationale behind the trends uncovered in these experiments. That said, the intro and/or discussion sections could be improved by placing this study’s findings in a broader context; what, for instance, is already known about larval diet requirements in mosquitoes, or insects in general? Nothing is mentioned of recent interesting findings on the requirement of bacteria for development (e.g., Coon et al., 2017; PMID: 28630299), and much more could be stated on similar diet studies in aegypti (e.g., Souza et al., 2019; PMID: 28630299) and Anopheles (numerous studies). As such, although the methodology and findings are sound and interesting in their own right, the unique contribution of the present study to the field of vector biology at large, or even to this particular line of inquiry within the field specifically, are not always readily apparent. However, this shortcoming may easily be addressed with a relatively short literature review in the introduction and especially the discussion, as suggested above.
  • Specific comments:
    1. Line 3: The phrase “Adult Outcomes” is somewhat vague. Perhaps consider modifying the word “Outcomes” for a more informative and impactful title.
    2. Line 26: Add “of” before “an” in the phrase “responsible for the transmission an…”
    3. Line 31: Please add a period at the end of this sentence.
    4. Line 67: Please provide company information (including city, state, country, if applicable) for MP Biomedicals in parenthesis. Also, should company information be provided for the RF? Was it pre-made?
    5. Line 70: After the sentence on AY composition, please consider adding a sentence briefly describing how RF was made (for example, by grinding rice into powder or mixing a pre-made powder with water, etc.).
    6. Lines 73-74: The diet combinations are listed in opposite order as presented later in the results (lines 119-120) and Fig. 1.
    7. Lines 74-78: Did the larvae eat all of the diet each day, or was there a fair amount of hold over that accumulated over the course of the feedings? A very brief comment in the text clarifying this may be helpful, as this relates to the issue of over-feeding and water fouling mentioned in line 78.
    8. Lines 75-77: The volumes go from presenting once decimal place to two, sometimes including a trailing zero, sometimes not. Please modify to remain consistent.
    9. Line 77: When you state “as well as during pupation,” do you mean that trays were given 0.15 mL of diet for each day the trays contained pupae (for 5 days, as mentioned in line 84?), starting at day 6 and beyond? Was this too feed residual 4th instars without over-feeding and creating foul water? Might be worth clarifying since the pupae obviously are not feeding.
    10. Lines 80-81: Why did you settle on the 0.4 larvae/mL density (also, 200 larvae/600 mL is , so I’m not sure this is best reported as “0.4 larvae per mL”)? We typically use 200 larvae/1,000 mL in our lab rearing, or 0.2 larvae per mL, and this seems to be an ideal for rearing medium body sized females (as in: Helinski and Harrington, 2011; PMID: 21485355). Is your density considered high density compared to other rearing protocols in the literature? Is it a “natural” density? Is that even known? Please consider citing precedent that you used to arrive upon and justify your choice rearing density.
    11. Lines 81-82: Rearing at 25°C is on the low end typical rearing temps (from my experience, usually 27-28°C). In fact, in line 90, 25°C is referred to as “room temperature.” Please justify, as this may alter developmental times, food intake, metabolism, etc., especially relative to other studies performed at higher rearing temperatures.
    12. Lines 85-86: Please explain why you subtracted the total number of larvae that reached pupation from the total number of starting larvae.
    13. Lines 113-124: All of this information is in the preceding methods section (lines 66-78); please consider shortening or eliminating altogether to reduce redundancy. Since Figure 1 presents no results, if possible, please place it after the paragraph containing the first reference to it in the text (so, after line 78).
    14. Figure 2: Is the “Time (days)” axis is really “Time from Onset of Pupation (days)”? These graphs show data from the “six days following initiation of pupation” (lines 130-131), correct?
    15. Line 135-136: Time to pupation of “~7 days” or “median of 9 days” refers to 7/9 days from hatching/first instar, correct? So then that would correspond to which day in Figures 2A and B? This is somewhat confusing since the X-axes of the graphs both end at day 6.
    16. Lines 136-140: There are no statistical tests or values listed in parenthesis in the text either here and elsewhere throughout the results to support these statements. Please add this information.
    17. Figure 2B: This was briefly mentioned in lines 129-132, but was this data analyzed statistically? If so, it would be helpful to share these results.
    18. Lines 155-168: The male data is presented first, then the female, in both paragraphs (both here and throughout the results), which is opposite of the flow of Fig. 2. Consider switching the order of presentation here and throughout the results to streamline correspondence between the text and figures.
    19. Lines 172-173: This does not appear to correspond with the figure: both 3P:1C and 1P:1C are labelled “a” with no indication of a statistically significant difference.
    20. Lines 176-178: 1P:1C males retain more glycogen than 0P:1C and 1P:3C males (this is clear), but don’t the 1P:1C males (denoted with “a”) also have more glycogen than the 3P:1C males (“b”)?
    21. Lines 178-179: 1P:1C males are only compared to 0P:1C males in this sentence, but 1P:1C males also differed from 1P:3C males, so why is only the former comparison singled out here?
    22. Line 179-181: The trend stated here for females appears to contradict a statement in line 175-176, so perhaps the former statement can be amended to reflect this nuance. Also, the authors state “In contrast to the trend observed for males, glycogen stores in adult female mosquitoes did not significantly vary according to diet (Figure 4b).” However, the graph really does appear to indicate that they do vary significantly. Without mentioning the overall ANOVA value, the reader is left wondering how close the test was to significant, especially given some of the near significant post-hoc comparisons.
    23. Lines 181-182: This is not a complete sentence; please combine with the preceding. Also, the authors state: “females fed the 1P:1C diet retained more glycogen stores than females fed the 0P:1C and 1P:3C diets (Figure 4b).” But doesn’t the 3P:1C group also retain more glycogen than both of these groups? If so, why isn’t that mentioned? If not, please explain the significance labels.
    24. Line 182-183: These lines indicate that the differences in the previous sentence were not statistically significant. The preceding sentence, then, seems misleading, as do the significance labels in the figure (I am assuming that different letters denote statistically significant differences). Please clarify what you are claiming and, overall, improve the quality of this entire paragraph; it is incomplete and difficult to follow.
    25. Lines 185-192: Now, female results are discussed first. Please do this, consistently, throughout.
    26. Lines 185-186: The authors state “Females fed the 0P:1C diet retained significantly more lipid stores than females fed either the 1P:1C or 3P:1C diets” but not, I take it, more than the 1P:3C diet? If not, then why is it also labelled “b”? Again, please explain, either in a figure legend or in the statistical analyses section of the materials and methods, what your labeling conventions are here for indicating statistically significant differences between groups.
    27. Lines 187-188: As before, the example cited here adds little to the results, and the authors have left out other relevant comparisons just mentioned. Please revise this.
    28. Lines 188-189: The authors state “Males fed the 0P:1C diet had a higher average amount of stored lipids per mosquito than males fed the 1P:1C diet” but isn’t that also true of the 0P:1C vs 3P:1C comparison? The lettering seems to indicate so.
    29. Lines 191-192: Again, why single out only the 1P:3C vs 1P:1C comparison?
    30. Lines 205-207: Please consider combining these two sentences for conciseness.
    31. Lines 207-208: The authors should speculate, briefly, as to why female body size was impacted to a greater extent by diet alterations compared to males.
    32. Lines 209-212: Consider combining these sentences with the previous paragraph.
    33. Lines 212-214: Consider combining these sentences with the paragraph that follows.
    34. Line 213: Please consider adding a comma after the phrase “with TOR signaling.”
    35. Lines 216-217: The sentence beginning with “As the TOR pathway” is incomplete; please complete it or merge it with the preceding sentence.
    36. Line 225: Add a citation after the sentence ending in “dFOXO,” or should the reader understand all of the references following the next sentence (line 227) as also applying to the preceding sentence? Consider splitting up these references.
    37. Line 233: Please consider replacing the word “emphasize” with a more suitable verb. The words “capacitating” (line 244) and “shunted” (line 248) are used in similar instances below and seem more fitting.
    38. Lines 238-239: Same issue as noted in “jj” above with line 255.
    39. Line 247: Please revise the informal language (“it’s possible”).
    40. Lines 256-257: Please provide a citation here showing that “the PI3K signaling cascade is activated when mosquito larvae are fed a diet with sufficient protein levels” as it appears that only Drosophila literature is cited above. However, if this is your hypothesis based off the Drosophila studies, rather than an allusion to established findings already known in mosquitoes, then please make that clear.
    41. Lines 269-271: Please provide citations.

Author Response

We thank the reviewers for the time and consideration given to this manuscript. Their comments were thoughtful, insightful and caught a number of issues that we had missed prior to submission. Our apologies for these oversights. We have addressed the comments to the best of our ability and we feel that the manuscript is much improved as a result. A couple of comments suggesting additional experimental work were not able to be addressed due to current limitations on lab work associated with the Covid-19 shutdown. However, these comments are well received and we plan to implement them in future analyses. Please find below our point by point responses to the reviewer’s comments.

Reviewer 1

Broad comments:

The materials and methods section would benefit from some further, relatively minor clarifications (see specific comments). The results would benefit from additional editing to improve readability and to clarify what conclusions the statistical methods and results do and do not support.

Thank you very much for your feedback. The materials and methods section has undergone significant revision to improve readability and provide additional information regarding all aspects of the work described. In particular, additional information regarding the nutritional and statistical analyses has been provided.

In general, the denoting of statistical significance in the graphs is confusing, particularly in the cases with double letter labels. Furthermore, some of the labels do not appear to correspond with the write up in the main text of the results. Few statistical test values are presented in the results (although they are often referred to), which is an omission that needs to be addressed.

We apologize for the confusion regarding the statistical annotations as well as for the accidental omission of the figure captions in the initial submission. Significant changes have been made to the materials and methods to clarify the statistical methods performed. The figure captions have been added and also include descriptions of the statistical annotations on the figures. We hope this clarifies the interpretation of the figures. We have also attempted to clarify the results to make the analysis and statistical interpretations easier to follow. We also discovered during our revision that the use of the parametric ANOVA analysis was inappropriate for this data as some of the data appears to violate normality. The statistical analyses have been updated using the non-parametric Kruskall-Wallis test followed by pairwise scoring using the Wilcoxon signed-rank test. The figures, captions and manuscript have been updated to reflect this change and observed significant differences between diets.

However, the discussion generally does an excellent job at teasing apart the implications and potential rationale behind the trends uncovered in these experiments. That said, the intro and/or discussion sections could be improved by placing this study’s findings in a broader context; what, for instance, is already known about larval diet requirements in mosquitoes, or insects in general? Nothing is mentioned of recent interesting findings on the requirement of bacteria for development (e.g., Coon et al., 2017; PMID: 28630299), and much more could be stated on similar diet studies in aegypti (e.g., Souza et al., 2019; PMID: 28630299) and Anopheles (numerous studies). As such, although the methodology and findings are sound and interesting in their own right, the unique contribution of the present study to the field of vector biology at large, or even to this particular line of inquiry within the field specifically, are not always readily apparent. However, this shortcoming may easily be addressed with a relatively short literature review in the introduction and especially the discussion, as suggested above.

Both the introduction and the results have been significantly revised in an attempt to provide additional context in which to fit our results and a more detailed review of related literature. We have made sure to integrate the findings of the Souza and Coon papers into the introduction (lines 67-70) and discussion sections (lines 319-333).

Specific comments:

  1. Line 3: The phrase “Adult Outcomes” is somewhat vague. Perhaps consider modifying the word “Outcomes” for a more informative and impactful title.

The title has been changed to “Impacts of Dietary Nutritional Composition on Larval Development and Adult Body Composition in the Yellow Fever Mosquito (Aedes aegypti)”

  1. Line 26: Add “of” before “an” in the phrase “responsible for the transmission an…”

“Of” has been added before “an.”

  1. Line 31: Please add a period at the end of this sentence.

A period has been added to line 31.

  1. Line 67: Please provide company information (including city, state, country, if applicable) for MP Biomedicals in parenthesis. Also, should company information be provided for the RF? Was it pre-made?

Company information for MP Biomedicals (Burlingame, CA, United States) was added to lines 79-80.  The rice flour is from the brand Erawan Group (Bangkok, Thailand). Company information for RF was added to line 80.

  1. Line 70: After the sentence on AY composition, please consider adding a sentence briefly describing how RF was made (for example, by grinding rice into powder or mixing a pre-made powder with water, etc.).

The RF is a pre-made powder, from the brand Erawan Group, which was added to the appropriate larval trays.

  1. Lines 73-74: The diet combinations are listed in opposite order as presented later in the results (lines 119-120) and Fig. 1.

Diet combinations presented in lines 77-79 now are listed in the same order as in figure 1 and in the results section.

  1. Lines 74-78: Did the larvae eat all of the diet each day, or was there a fair amount of hold over that accumulated over the course of the feedings? A very brief comment in the text clarifying this may be helpful, as this relates to the issue of over-feeding and water fouling mentioned in line 78.

Based on visual inspection of the trays the larvae ate most if not all of the diet each day. This observation has been added to line 95.

  1. Lines 75-77: The volumes go from presenting once decimal place to two, sometimes including a trailing zero, sometimes not. Please modify to remain consistent.

The volumes mentioned in lines 90-93 now present two decimal places.

  1. Line 77: When you state “as well as during pupation,” do you mean that trays were given 0.15 mL of diet for each day the trays contained pupae (for 5 days, as mentioned in line 84?), starting at day 6 and beyond? Was this too feed residual 4th instars without over-feeding and creating foul water? Might be worth clarifying since the pupae obviously are not feeding.

Trays were given 0.15mL of diet for each day that the trays countained pupae in order to feed residual fourth instars and to avoid fouling the water. This has been added to the text in lines 93-95.

  1. Lines 80-81: Why did you settle on the 0.4 larvae/mL density (also, 200 larvae/600 mL is , so I’m not sure this is best reported as “0.4 larvae per mL”)? We typically use 200 larvae/1,000 mL in our lab rearing, or 0.2 larvae per mL, and this seems to be an ideal for rearing medium body sized females (as in: Helinski and Harrington, 2011; PMID: 21485355). Is your density considered high density compared to other rearing protocols in the literature? Is it a “natural” density? Is that even known? Please consider citing precedent that you used to arrive upon and justify your choice rearing density.

Larval density is an important consideration in the context of larval development. We have removed “0.4 larvae per mL” for the erroneous math, and chose to only include the description 200 larvae in 600 mL as we feel this phrasing is more useful to readers. When considering larval rearing density, we assume that larvae per water surface area (cm2 ) is a more important metric than density (cm3), if adequate swimming depth is maintained (>0.5cm water). By this metric our density of 0.2 larvae/cm2  is between the large body density used for males and the medium body density used for females (0.13 and 0.37 larvae/cm2)  in Helinski and Harrington, 2011; PMID: 21485355, which cites Ponlawat and Harrington, 2007; PMID: 17547226.  We have included the measurements and product information for the bioquip larval rearing trays that we use, which are relatively shallow compared to the trays used in the citations above. We have found the volume used in our experiments (600mls) nicely fills these trays with little need for refilling. Additional water is added to maintain depth, and this information has been added to the manuscript (line 107).

  1. Lines 81-82: Rearing at 25°C is on the low end typical rearing temps (from my experience, usually 27-28°C). In fact, in line 90, 25°C is referred to as “room temperature.” Please justify, as this may alter developmental times, food intake, metabolism, etc., especially relative to other studies performed at higher rearing temperatures.

The manuscript has been revised to accurately reflect our rearing conditions of 26°C with a RH of 80%. This temperature was described in Clemmons et al., 2010 PMID: 20889704. We also maintain Tsetse flies in the same chamber as our Aedes aegypti, which is why our rearing chamber is kept at an ambient temperature slightly cooler than 27-28 degrees Celsius as both species seem to function well at this temperature. The temperature in line 90 was also erroneous and has been corrected, we stored the dried mosquitoes in our lab at room temperature, which is typically 20-21 degrees celsius, not 25.

  1. Lines 85-86: Please explain why you subtracted the total number of larvae that reached pupation from the total number of starting larvae.

The total number of larvae that reached pupation was subtracted from the total number of starting larvae in order to serve as a proxy for the percent of the original population that survived to pupation.

  1. Lines 113-124: All of this information is in the preceding methods section (lines 66-78); please consider shortening or eliminating altogether to reduce redundancy. Since Figure 1 presents no results, if possible, please place it after the paragraph containing the first reference to it in the text (so, after line 78).

This paragraph has been removed from from the text.

  1. Figure 2: Is the “Time (days)” axis is really “Time from Onset of Pupation (days)”? These graphs show data from the “six days following initiation of pupation” (lines 130-131), correct?

That is correct and the graph axis of Figure 2 have been modified to reflect this.

  1. Line 135-136: Time to pupation of “~7 days” or “median of 9 days” refers to 7/9 days from hatching/first instar, correct? So then that would correspond to which day in Figures 2A and B? This is somewhat confusing since the X-axes of the graphs both end at day 6. Figure captions have been added in order to explain the labeling scheme.
  2. Lines 136-140: There are no statistical tests or values listed in parenthesis in the text either here and elsewhere throughout the results to support these statements. Please add this information.

We apologize for the omission of figure captions which were supposed to be included with the initial submission but missed due to an oversight. The captions have been added which explain the statistical analyses performed as well as how the labeling scheme represents significant differences between the treatments. We have also included the R code and data sheets used to generate the figures and statistics.

  1. Figure 2B: This was briefly mentioned in lines 129-132, but was this data analyzed statistically? If so, it would be helpful to share these results.

The data was analyzed statistically by Kaplan meier survival analysis followed by pairwise scoring by log-rank test. The materials and methods have been updated to provide the details of the statistical analyses. The figure captions have been added and include descriptions of the statistical analyses and significance annotations..

  1. Lines 155-168: The male data is presented first, then the female, in both paragraphs (both here and throughout the results), which is opposite of the flow of Fig. 2. Consider switching the order of presentation here and throughout the results to streamline correspondence between the text and figures.

For clarity the text has been modified so females are discussed first, then males to follow the figure presentation (lines 260-271)

  1. Lines 172-173: This does not appear to correspond with the figure: both 3P:1C and 1P:1C are labelled “a” with no indication of a statistically significant difference.

This was due to an incorrect interpretation on our part. The results section has been heavily revised to reflect the results and p-values have been added to the text for each comparison.

  1. Lines 176-178: 1P:1C males retain more glycogen than 0P:1C and 1P:3C males (this is clear), but don’t the 1P:1C males (denoted with “a”) also have more glycogen than the 3P:1C males (“b”)?

Thank you for identifying this. The results section has been amended to correctly interpret the results and include p-values for the comparisons described.

  1. Lines 178-179: 1P:1C males are only compared to 0P:1C males in this sentence, but 1P:1C males also differed from 1P:3C males, so why is only the former comparison singled out here?

The comparison between glycogen stores of Medium protein diet (was 1P:1C) males and low protein (was 0P:1C) males is singled out since these two groups had the largest difference in average glycogen stores. A statistically significant difference was also observed when comparing 1P:1C males to 1P:3C males, but the difference was not to the extent to that seen when comparing males fed 1P:1C versus a lower protein diet (0P:1C). The following sentence was added to the text in line 222: Furthermore, on average, 1P:1C males retain 35.13 ug of glycogen more than 1P:3C males.

  1. Line 179-181: The trend stated here for females appears to contradict a statement in line 175-176, so perhaps the former statement can be amended to reflect this nuance. Also, the authors state “In contrast to the trend observed for males, glycogen stores in adult female mosquitoes did not significantly vary according to diet (Figure 4b).” However, the graph really does appear to indicate that they do vary significantly. Without mentioning the overall ANOVA value, the reader is left wondering how close the test was to significant, especially given some of the near significant post-hoc comparisons.

This was due to an incorrect interpretation on our part. The results section has been heavily revised to reflect the results and p-values have been added to the text for each comparison.

  1. Lines 181-182: This is not a complete sentence; please combine with the preceding. Also, the authors state: “females fed the 1P:1C diet retained more glycogen stores than females fed the 0P:1C and 1P:3C diets (Figure 4b).” But doesn’t the 3P:1C group also retain more glycogen than both of these groups? If so, why isn’t that mentioned? If not, please explain the significance labels.

This was an oversight on our part. The results section has been revised to reflect the significant differences in glycogen levels between the two higher protein groups relative to the lower protein groups and p-values have been added to the text for each comparison.

  1. Line 182-183: These lines indicate that the differences in the previous sentence were not statistically significant. The preceding sentence, then, seems misleading, as do the significance labels in the figure (I am assuming that different letters denote statistically significant differences). Please clarify what you are claiming and, overall, improve the quality of this entire paragraph; it is incomplete and difficult to follow.

The results section has been heavily revised to reflect the results and p-values have been added to the text for each comparison. These lines were re-ordered for clarity to emphasize the fact that the observed lipid results for females may reveal a statistically significant trend in future experiments with a larger sample size.

  1. Lines 185-192: Now, female results are discussed first. Please do this, consistently, throughout.

We apologize for the inconsistency in the presentation of the results. All the nutritional results have been rewritten for clarity and consistently order the presentation of the results such that females are described first followed by the results from males.

  1. Lines 185-186: The authors state “Females fed the 0P:1C diet retained significantly more lipid stores than females fed either the 1P:1C or 3P:1C diets” but not, I take it, more than the 1P:3C diet? If not, then why is it also labelled “b”? Again, please explain, either in a figure legend or in the statistical analyses section of the materials and methods, what your labeling conventions are here for indicating statistically significant differences between groups.

The results section has been heavily revised to accurately reflect the significant differences in the results and p-values have been added to the text for each comparison. The figure captions and materials and methods have been updated to clearly describe the statistical analyses and significance annotations.

  1. Lines 187-188: As before, the example cited here adds little to the results, and the authors have left out other relevant comparisons just mentioned. Please revise this.

This section has been revised as described in the previous section.

  1. Lines 188-189: The authors state “Males fed the 0P:1C diet had a higher average amount of stored lipids per mosquito than males fed the 1P:1C diet” but isn’t that also true of the 0P:1C vs 3P:1C comparison? The lettering seems to indicate so.

The text at lines 227-228 was amended to read, “Males fed the 0P:1C diet had a higher average amount of stored lipids per mosquito than males fed the 1P:1C or 3P:1C diets.”

  1. Lines 191-192: Again, why single out only the 1P:3C vs 1P:1C comparison?

We have removed the 1P:3C (Med-low) vs 1P:1C (Med) comparison was deleted from the text.

  1. Lines 205-207: Please consider combining these two sentences for conciseness.

The two sentences in lines 213-214 were combined with “and.”

  1. Lines 207-208: The authors should speculate, briefly, as to why female body size was impacted to a greater extent by diet alterations compared to males.

Similar results have been noted in work by Andersen et al., 2010  PMID: 19931279 in Drosophila melanogaster, and we have added our brief speculation in lines 265-266.

  1. Lines 209-212: Consider combining these sentences with the previous paragraph.

Lines 266-269 were combined with the previous paragraph.

  1. Lines 212-214: Consider combining these sentences with the paragraph that follows. Lines 270-272 were combined with the paragraph that follows.
  2. Line 213: Please consider adding a comma after the phrase “with TOR signaling.”

A comma has been added to line 222.

  1. Lines 216-217: The sentence beginning with “As the TOR pathway” is incomplete; please complete it or merge it with the preceding sentence.

The sentence beginning with “As the TOR pathway” has been merged with the following sentence in line 217.

  1. Line 225: Add a citation after the sentence ending in “dFOXO,” or should the reader understand all of the references following the next sentence (line 227) as also applying to the preceding sentence? Consider splitting up these references.

The references have been split to associate them more closely with the relevant statements.

  1. Line 233: Please consider replacing the word “emphasize” with a more suitable verb. The words “capacitating” (line 244) and “shunted” (line 248) are used in similar instances below and seem more fitting.

“Emphasize” has been replaced with “capacitate” in line 242.

  1. Lines 238-239: Same issue as noted in “jj” above with line 255.

The references have been split to associate them more closely with the relevant statements.

  1. Line 247: Please revise the informal language (“it’s possible”).

“It’s possible” has been replaced with “One potential hypothesis is that [...]” in lines 268-269.

  1. Lines 256-257: Please provide a citation here showing that “the PI3K signaling cascade is activated when mosquito larvae are fed a diet with sufficient protein levels” as it appears that only Drosophila literature is cited above. However, if this is your hypothesis based off the Drosophila studies, rather than an allusion to established findings already known in mosquitoes, then please make that clear.

This is our hypothesis based on Drosophila literature. “We hypothesize that [...]” was added to clarify that this statement is based on Drosophila studies.

  1. Lines 269-271: Please provide citation.

The citation for the Dias, Rodrigues, and Silva paper, “Effect of acute and chronic exposure to ammonia on different larval instars of An. darlingi,” was added to the text.

Reviewer 2 Report

The research article 'Impacts of dietary nutritional composition on larval development and adult outcomes in the yellow fever mosquito (Aedes aegypti)’ by van Schoor et al., investigates the role that nutrition plays in the development of adult Ae. aegypti. The authors supply larvae with diets consisting of various proportions of carbohydrates/ protein and study the effect this has on time to pupation time, wingspan, body weight/ size and nutritional stores of sugars/lipids. This paper is of interest to the community as it illustrates nutritional requirements needed to generate healthy, lab reared Ae. aegypti.   

There are a few points that the authors should address in order to improve their manuscript.

General comments:

General proof reading is required to improve the English language, style and grammar used. For example, 

Line 11- effects nutrition HAS on growth

Line 23- Ae. aegypti in italics 

Line 26- for the transmission OF an array 

Line 30- Ae. aegypti (small a for aegypti) etc 

Some comments are quite casual for a scientific article e.g. 

Line 262- as to what could be happening here

Line 282- that they are dying as a result 

Check consistency of formatting throughout. E.g. Figure 2  x-axis Time (days) or Time(Days)/ spacing between value and units (section 2.4)

None of the figures are accompanied by a figure legend 

Supplemental data is not cited within the MS 

 Specific comments:

I suggest that the reviewers mention in the abstract/ introduction that this information relates to the diet of laboratory reared mosquitoes. 

The authors mention TOR/PI3K pathway involvement in physiological processes however they do not directly relate it to the MS findings. E.g. Line 255- 'the 1P:3C diet provides enough protein to stimulate the TOR pathway to induce cellular growth, replication and organismal development' How do the authors know this is this the case in this instance?    

Reference 20 appears to be incomplete in the reference list. I suggest using a more appropriate citation than it in lines 208 and 252.    

It would add interest if the authors were to demonstrate how these diets affect blood feeding in females (as mentioned in lines 54-57) 

Line 73- assume that P is protein and C is carbohydrate 

I found the description of the proportions of P/C confusing and misleading without looking at Figure 1. E.g. 1P:0C I would assume would mean no carbohydrate however it contains 30% C/  3P:1C seems closer to 1:1 ratio than 1P:1C is.   

Line 100- I assume 16,17 and 18 refers to references? 

Line 208- Can the authors suggest why this would be the case?

Line 287- From the results of the MS, what proportion of P/C do the authors suggest should be used to rear Ae. aegypi successfully?   

Author Response

We thank the reviewers for the time and consideration given to this manuscript. Their comments were thoughtful, insightful and caught a number of issues that we had missed prior to submission. Our apologies for these oversights. We have addressed the comments to the best of our ability and we feel that the manuscript is much improved as a result. A couple of comments suggesting additional experimental work were not able to be addressed due to current limitations on lab work associated with the Covid-19 shutdown. However, these comments are well received and we plan to implement them in future analyses. Please find below our point by point responses to the reviewer’s comments.

Reviewer 2

General comments:

General proof reading is required to improve the English language, style and grammar used. For example,

  1. Line 11- effects nutrition HAS on growth

“Has” was added.

  1. Line 23- aegypti in italics

Ae. aegypti was italicized.

  1. Line 26- for the transmission OF an array

“Of” was added.

  1. Line 30- Ae. aegypti (small a for aegypti) etc

The “a” in “aegypti” was changed to lowercase.

  1. Some comments are quite casual for a scientific article e.g. Line 262- as to what could be happening here

Overall the manuscript has been edited to eliminate casual language. “As to what could be happening here” was changed to “there are a few hypotheses that can explain this result” in lines 337-338.

Line 282- that they are dying as a result

“That they are dying as a result” was changed to “they are generating a toxic amount of nitrogenous waste” in line 362.

  1. Check consistency of formatting throughout. E.g. Figure 2 x-axis Time (days) or Time(Days)/ spacing between value and units (section 2.4)

Thank you for this comment, the axis of figure 2 have been modified to maintain consistency and be more descriptive.

  1. None of the figures are accompanied by a figure legend.

We apologize for this oversight. Figure legends have been added for all figures and include descriptions of statistical analyses and annotations.

  1. Supplemental data is not cited within the MS

We had thought the supplemental data files had been included with the prior submission however they must have been inadvertently excluded. The revision includes the R script used for data analysis as well as the raw data files used to generate the figures and statistical analyses.

Specific comments:

  1. I suggest that the reviewers mention in the abstract/ introduction that this information relates to the diet of laboratory reared mosquitoes.

The mosquitoes are of the laboratory-raised Rockefeller strain. This has been added to lines 12-13.

  1. The authors mention TOR/PI3K pathway involvement in physiological processes however they do not directly relate it to the MS findings. E.g. Line 255- 'the 1P:3C diet provides enough protein to stimulate the TOR pathway to induce cellular growth, replication and organismal development' How do the authors know this is this the case in this instance?

Thank you for this comment, this is our hypothesis, however it is not directly tested in these experiments. The language of the manuscript has been updated to reflect this.

  1. Reference 20 appears to be incomplete in the reference list. I suggest using a more appropriate citation than it in lines 208 and 252.

Thank you for catching this. The references have been updated to ensure the appropriate references are cited in the text.

  1. It would add interest if the authors were to demonstrate how these diets affect blood feeding in females (as mentioned in lines 54-57)

This is definitely of interest to us. However, at this time our capacity to perform additional experiments is limited due to restrictions associated with COVID-19 isolation policies. We are planning on investigating differences in blood feeding and sugar feeding behaviors as well as female fecundity on the diets presented here.

  1. Line 73- assume that P is protein and C is carbohydrate.

The diet names have been modified to (Low, Med-low, Med, Med-high and High) for easier interpretation and clarity throughout the manuscript. The new diet names refer to the relative protein level of the diet (Figure 1). 

  1. I found the description of the proportions of P/C confusing and misleading without looking at Figure 1. E.g. 1P:0C I would assume would mean no carbohydrate however it contains 30% C/ 3P:1C seems closer to 1:1 ratio than 1P:1C is.

We agree with the reviewer and have modified the diet names as noted above. Figure 1 has also been labelled with the percentages of nutrients within each diet.  

  1. Line 100- I assume 16,17 and 18 refers to references?

References 16, 17, and 18 were erroneously cited here. This has been corrected and the proper reference included.

  1. Line 208- Can the authors suggest why this would be the case?

This section has been amended with observations from Drosophila indicating that female body size may be more intensely impacted by nutritional variation in larval diets due to the increased nutritional demands associated with metabolic preparations for nutrient intensive reproductive processes such as vitellogenesis and oogenesis.

  1. Line 287- From the results of the MS, what proportion of P/C do the authors suggest should be used to rear Ae. aegypi successfully?

Based on our results, larvae fed the 1P:3C (now the Med-low) diet exhibited the most preferential life history traits in terms of body size and nutritional stores. However, this diet might not be optimal as not all combinations of protein to carbohydrate ratios have been tested and there may be other factors not accounted for in these experiments that impact growth. This is reflected in the revised discussion.

Reviewer 3 Report

The manuscript brings topics of high interest, specially considering the development and investment on alternative technologies besides insecticide application.

Overall, I think the manuscript can be improved on material and methods section, as for the current objectives.

As a matter to discuss the development of a diet for regardless its application, I think the paper has very interesting points. However to describe mosquitoes nutrition needs without properly isolating the nutrients to test them and having a great variety of them in each diet, it is quite difficult to correlate the results to the conclusion, only with the presented results. 

I think some additional experiments/results are needed, in order to confirm the conclusions and discussion.

There is no mention about the strain origin and for how long its kept in insectary, same for the maintenance procedure of this strain (if any). Only the strain selection pressure can be a critical factor, which have been favouring the selection of specific conditions, and in face of a new diet has a different behaviour.

Also the use a standard diet. The daily rearing condition of this strain might already show a pattern of fitness for this strain and this is completely ignored.

In your R code script, the line #314, the object "FWing_Sig_Fz", should instead be: "FWing_Sig_F"? I am also concerned with the tests used and it would be nice to prior to the models selected, to evaluate whether a transformation/normalization should be used for the selected linear model, or if a non-parametric model would be better.

Author Response

We thank the reviewers for the time and consideration given to this manuscript. Their comments were thoughtful, insightful and caught a number of issues that we had missed prior to submission. Our apologies for these oversights. We have addressed the comments to the best of our ability and we feel that the manuscript is much improved as a result. A couple of comments suggesting additional experimental work were not able to be addressed due to current limitations on lab work associated with the Covid-19 shutdown. However, these comments are well received and we plan to implement them in future analyses. Please find below our point by point responses to the reviewer’s comments.

Reviewer 3

Comments and Suggestions for Authors

  1. Overall, I think the manuscript can be improved on material and methods section, as for the current objectives.

The materials and methods section (along with most of the other sections) has undergone a significant rewrite to include additional information regarding the methods and statistical analyses performed.

As a matter to discuss the development of a diet for regardless its application, I think the paper has very interesting points. However to describe mosquitoes nutrition needs without properly isolating the nutrients to test them and having a great variety of them in each diet, it is quite difficult to correlate the results to the conclusion, only with the presented results. I think some additional experiments/results are needed, in order to confirm the conclusions and discussion.  

There is no mention about the strain origin and for how long its kept in insectary, same for the maintenance procedure of this strain (if any). Also the use a standard diet. The daily rearing condition of this strain might already show a pattern of fitness for this strain and this is completely ignored. Only the strain selection pressure can be a critical factor, which have been favouring the selection of specific conditions, and in face of a new diet has a different behaviour.

The lack of detailed breakdown of the exact compounds within our dietary formulations is a significant limitation of our current study as we don’t know the exact breakdown of all the potential compounds found within the autolyzed yeast and rice powder. There are definitely other factors included in the diet predominantly originating from the autolyzed yeast, however a detailed metabolomic breakdown of the diets are somewhat beyond the scope of this study.  However, we believe our estimates of the relative proportions of carbohydrates and protein, while relatively crude, are accurate on the whole.

The mosquito strain used is a very important consideration and we have added our strain information and standard larval rearing protocol to the methods section 1.2. We use the Rockefeller strain, a highly inbred lab strain that has been in culture for decades. The benefit of this strain is that it has been used in many other physiological studies which facilitates comparison with results from other research groups. Our main colony is reared in the lab using a diet of tetramin fish food.

To address these issues we have added some material on these topics in the discussion:

“One consideration not explored in this study is the adaptability of our lab strain, which has been maintained on a diet of Tetramin fish food. It is possible that, in the presence of sustained selective pressure,that adaptation to this diet could influence the response to the diets used in this work. It is also important to note that while autolyzed Brewer’s yeast extract is high in protein it also includes various other nutritional components such as lipids, vitamins, nucleic acids that are not cannot be accounted for by these experiments and likely play important roles in growth and development, but most material is proteinaceous [21,22]. Future studies are required to determine the role that these compounds play in the effects observed here.”.

We agree with the reviewer that conducting follow up experiments to specifically examine the role of key nutrients in the regulation of the discussed developmental pathways is important and something which we are planning on performing. Unfortunately, due to current COVID-19 restrictions on laboratory work, additional experiments cannot currently be conducted expeditiously.

  1. In your R code script, the line #314, the object "FWing_Sig_Fz", should instead be: "FWing_Sig_F"? I am also concerned with the tests used and it would be nice to prior to the models selected, to evaluate whether a transformation/normalization should be used for the selected linear model, or if a non-parametric model would be better.

Thank you for catching this! We had forgotten to check for normality prior to analysis of the body size and nutritional data. We tested the data for normality using the Shapiro-Wilks test and found that it was likely to be non-normal. As a result, the data has been reanalyzed using the non-parametric Kruskal-Wallis Test followed by pairwise comparison with the Wilcoxon signed-rank test. The figures, figure captions amd materials and methods have been updated accordingly. Statistical comparison of developmental rates was performed by Kaplan-meier survival curve analysis followed by Log-Rank pairwise comparisons. As this is a non-parametric test of binomial data it does not require transformation or normalization of the data.

  1. Line 43 This is the commercial name. Please change to a more general one, or include specific name for the other food type.

The text has been modified to specify Tetramin, as it is particularly common mosquito feed, while noting that it is a brand and that other varieties of fish food can be used.

  1. Line 50 Please define as it was done for the PI3K. I suppose you mean "target of rapamycin"- The acronym has been defined.
  2. Why this specific two types? other diets (even those mentioned in introduction as common in different) were tested?

Autolyzed yeast was chosen as it contains a high amount of protein relative to its carbohydrate composition. Rice powder was chosen due to its high relative amount of carbohydrates. The high relative protein/carbohydrate compositions of these ingredients facilitated the production of dietary blends which allowed us to tailor the relative proportions of these basic nutrients across the different diets. In addition, both of these ingredients are readily available and inexpensive.

  1. Line 74  ——     1. Were these larvae synchronized for checking developmental time?

All  larvae were synchronized at hatching, so they should all be of an equivalent physiological state.

  1. Any argument about the consumption of the remaining yolk in the L1 gut, if not synchronize and the addition of diet at this day?.
  2. Why this specific amount of diet on this day?

This amount of diet used was based on the amounts used during our standard colony larval rearing methods using Tetramin fish food. We expected this volume of food would be appropriate to provide adequate nutrition while reducing the risk of fouling the water to prevent the trays from becoming rancid.

  1. The amount (including other days) was per day of feeding, without correction for the total amount of larvae?

Total larvae were controlled in all trays at 200 larvae per tray. We assumed that larval survivorship is comparable between the 4 replicate trays.

  1. Why these quantities were selected for each feeding day?

This rationale is described in response to comment 8.3

  1. Line 78  ——   1. Was the water ever changed during the feeding?

Trays were initially started with 600mL of water. Water was added to each tray to maintain a constant volume throughout the experiment, this information was added to the methods section 2.2

  1. Are there any observation regarding whether the total consumption of diet was achieved?

Thank you for this comment. Based on visual inspection larvae consumed most if not all of the diet provided, this has been clarified in the materials and methods section 2.1.

  1. Line 80- from each strain? there is any information about the strain and the standard maintenance of this strain? If not should I assume that this were recently field collection? and in this case, where it was collected, which trapping system?

This has been addressed above.

  1. Line 81- I guess this was the room temperature, and not the water, is this correct?

The mosquitoes are maintained in a walk-in environmental chamber held at 26 degrees Celsius.

  1. Line 98- how many replicates? By the data provided, is it correct to say 3 replicates?

Yes, there were three replicates per dietary group.

  1. Line 107- should also include the version of R used under RStudio, which is even more important the the RStudio version.

Thank you for this comment. R version 3.6.2 was used, as the manuscript has been updated so this is now indicated in the statistical analysis section of the materials and methods.

  1. Line 107- Was the data distribution checked for normality before selecting the linear model and ANOVA? Was the data transformed to fit a Gaussian curve?

This is addressed above.

  1. Line 110- The code provided has many more packages and functions, would be nice to also include those with their respective citation.

Thank you for this comment. All packages in the code (ggfortify, survminer, ggplot2, ranger, forcats, dplyr, ggthemes, ggpubr, and aod) have been annotated with the associated citations.

  1. Line 111- As you provided the code, there is no need to place the functions here. I would rather mention the existence of the script file saying that all packages used can be found there, instead of the functions.

References to the methods and functions have been removed from the materials and methods

  1. Line 129-130- This is an important information. Because the diet contains some carbohydrate and lipids (and this last one, more than RF only). I would expect the larvae to take longer to develop. Is there any repetition of this experiment? how it behaved among the replicates?

Three biological replicates were included in the form of three trays per diet. The larval growth rates between trays within diets was fairly consistent..

  1. Line 142- Please include the legend for the figure and describe the legend (if possible to place maybe at the bottom, once is a common legend I guess).

Our apologies for the accidental omission of the figure captions and legends. Captions have been added to the figures and legends defining the diets can be found within the figures.

  1. Also standardize the axis and their legends. For example in A the y axis the numbers are different from y axis in B. also the spacing between words on the x axis.

The figures have been modified so the axis is standardized between A and B.

  1. I would also recommend to find a different color palette, some colors are quite similar and in A is very difficult to see the different lines and points, specially between 1P:3C and 1P:1C.

Thank you for this comment, we agree with the reviewer and the figures have been modified so as to include a more distinct color palette.

  1. Line 170- Please insert the legend of the figure. Also use one unit throughout the manuscript (mg or ug).

Units have been reformatted and standardized to µg in the text and on the figures.

  1. Line 173- keep consistent the unit the symbols are different along the manuscript, and this one is more appropriate than "ug".

The axis of figure 2 has been modified to use the unit µg and the use of µ is standardized throughout the manuscript.

  1. Line 194- Please include the figure legend. standard the figure layout (white/gray background). Figure captions have now been uploaded, and we have modified the figure so that the layout is consistent between figures.

  1. Figure A. Is the male sugar concentration really not significant? is there any outlier in the sample (usually the boxplot already point out as a point what could be an outlier)

The data for the male sugar plot (in particular the 75AY:25RF diet - renamed Med-high) had a large amount of variability with 4 points showing higher levels of sugar while the remaining 10 were at levels equivalent to those observed in the other diets. As it was more than just one or two points the higher values were not treated as outliers in the boxplot. We were concerned about dropping more than one or two values as outliers, so they were left in the dataset. The high level of variance associated with this set is likely why it is not significantly different from the other diets in the male dataset. I suspect the high readings are likely outliers as the majority of the dataset is very similar to sugar values observed in the other diets.

Figure B. same for females for glycogen concentration.

After reanalysis using a non-parametric statistical approach due to the non-normality of the data, the glycogen stores for each diet are now significantly different from one another. Thank you for catching this as use of the Kruskal-Wallis and Wilcoxon tests appears to have improved the accuracy of determination statistical significance between groups.

  1. Line 208- I don't think this is the reference to corroborate your findings.

Thank you for catching this. This reference was misplaced due to an issue with our reference management software. The citation has been updated to reflect the correct publication.

  1. Line 209-211- No evaluation or correction was done regarding the role of lipids on both diets (their quantities and type). What would be the impact of diets without lipids or these same diets in absence of lipids or equivalent quantities and type? We agree that lipid levels within diets may play a role in development. The lipids within the diets used for this work are associated with the autolyzed yeast component of the diets and make up a relatively low proportion of the diet (ranging from 1% to 9%) based on the nutritional data derived from the yeast extract composition. Unfortunately, we do not know the exact types of lipids included in the yeast extract. Most are likely associated with the yeast membrane structures and or are bound by lipid binding proteins. Manipulating lipids in a mosquito diet is somewhat difficult as increased lipid content makes solubility of the diet go down. Also increased amounts of lipids can disrupt the water surface tension which prevents the larval mosquitoes from breathing properly. Carbohydrates and proteins are much more water soluble and likely to be found in higher abundances in natural settings so we feel that these are more likely to constitute the majority of the mosquitoes diet. However, the role of lipids in larval development is a very interesting question and something we will address in the future.

  1. Line 214- Again, I disagree about the use of this specific reference for this statement. Please include an appropriate reference.

Thank you for catching this. This reference was misplaced due to an issue with our reference management software. The citation has been updated to reflect the correct publication.

  1. 231-233- The combination of different proportions of the diet still have a great amount of carbohydrate. the AY diet has 30% carbohydrate and contains more lipids too. I already mentioned this issue with the lipid content of each diet that is not equivalent and it is unknown its properties.

Thank you for this comment, the diet names have been modified for clarity throughout the manuscript; the new diet names refer to the relative protein level of the diet (Figure 1).  We agree with the reviewer that conducting experiments on the role of lipids in larval nutrition would be important regarding development time. Unfortunately, due to COVID-19 restrictions on laboratory work, additional experiments cannot currently be conducted expeditiously.

  1. This sounds as speculation, because there could be other diet components, and they were not explored.

We know rough proportions of carbs and protein- other factors are there, but unknown. Added in lines 498-502 of the discussion: “It is also important to note that while autolyzed Brewer’s yeast extract is high in protein it also includes various other nutritional components such as lipids, vitamins, nucleic acids that are not cannot be accounted for by these experiments and likely play important roles in growth and development, but most material is proteinaceous [21,22]. Future studies are required to determine the role that these compounds play in the effects observed here.”

  1. Line 245-247- Insulin levels were not measured. so the sentence is just speculating over something without literature reference and experimentation.

We have amended lines 563-566 highlight that these statements represent speculation of possible mechanisms underlying the observed results (“We speculate that [...], “we hypothesize that [...]”).

  1. Line 252- I don't think this is the suitable reference.

This reference was misplaced due to an issue with our reference management software. The citation has been updated to reflect the correct publication.

  1. Line 262-263- So, by this statement I would guess that the water wasn't change during the experiment, including beginning of pupation. How was the water aspect? It would be interesting to see whether water change would impact the results.

Initially, 600mL of water was added to each tray. Water was added to each tray on a daily basis to compensate for evaporation. This is now reflected in the text in lines 106-107. Biochemical analysis of the water over time for the different diets would be very interesting and informative as to the types of waste products produced by the larvae on different dietary regimens. This is something we will consider for future experimental work.

  1. Line 280-281- Again speculation, at least for me, that had no further information about this experimental cluster.

The wording in lines 612-614 was changed to reflect that this comment is speculation: “We speculate that […]”. In future experiments, we are trying to find the threshold of ammonia that becomes toxic to Aedes aegypti larvae.

  1. It is also known that this mosquito species filters the water and microorganism could also be the source of energy and nutrients. So there should be a balance offered by the chosen diet to impact the proliferation of certain microorganism, this can also be related to the gut microbiota of the insect.

This is definitely an important consideration. We have integrated commentary on and references to papers on the impact of microbiota on larval mosquito development in the introduction and discussion. Microbiome work on the larval water and resulting adults from the different diets are also things we are interested in doing in the future.

  1. Line 290- Is this the most recent publication on this subject?

We felt that this chapter provides a particularly relevant and informative overview of this dynamic, however we have also added a more up to date reference which covers the topic of regarding the evolution and diversity of nutrient sources adapted to by insects.

  1. Line 292- Thinking on this natural habitat, what would you think it would be the more predominant molecule that would compose the diet, protein? carbohydrate? lipids? and in artificial containers?

Both natural and artificial container habitats can be composed of variable compositions including leaf detritus, insect detritus and microbial communities. The protein/lipid/carbohydrate composition would depend on the contents of detritus, but nutrient composition is rarely reported in these terms in modern studies, but instead as Carbon and Nitrogen ratios. Previous work indicates that larval environments are often relatively rich in Nitrogen, and we assume protein. The manuscript has been updated at lines to include references to natural habitat composition. It would be very interesting to analyze the protein/lipid/carbohydrate composition of various natural larval habitats, but we feel this is outside of the scope of the study presented here.

  1. Line 292- Have evolved to- and environmental pressure selected more adapted populations to develop"

Thank you, this line has been edited to use more appropriate language.

  1. Line 299- reproduction- egg development. females don't use blood for reproduction. They can even have a blood meal and lay few eggs without previously mating.

Thank you for this comment, the manuscript has been edited at line 592 to say “egg development”.

Round 2

Reviewer 2 Report

The authors have adequately addressed my concerns. However, a thorough proof read is still required to ensure consistency throughout. 

Reviewer 3 Report

Thanks for the revised manuscript. Indeed those changes, along with the other reviewers made the manuscript more robust.

I only would like to mention that scientific names (specially non-Aedes aegypti ones) should be revised for appropriate formatting in italics.